# Transmission of *Xylella fastidiosa* Subspecies *Pauca* Sequence Type 53 by Different Insect Species

**DOI:** 10.3390/insects10100324

**Published:** 2019-09-29

**Authors:** Vincenzo Cavalieri, Giuseppe Altamura, Giulio Fumarola, Michele di Carolo, Maria Saponari, Daniele Cornara, Domenico Bosco, Crescenza Dongiovanni

**Affiliations:** 1Istituto per la Protezione Sostenibile delle Piante, CNR, 70126 Bari, Italy; 2Centro di Ricerca, Formazione e Sperimentazione in Agricoltura “Basile Caramia” (CRSFA), 70010 Locorotondo (Bari), Italy; 3Instituto de Ciencias Agrarias, Consejo Superior de Investigaciones Cientificas, ICA-CSIC, 28006 Madrid, Spain; 4Dipartimento di Scienze Agrarie, Forestali e Alimentari, Università degli Studi di Torino, 10095 Grugliasco, Italy

**Keywords:** bacterium, cherry, olive, almond, myrtle –leaf milkwort, periwinkle, spittlebugs, vectors

## Abstract

Diseases associated with *Xylella fastidiosa* have been described mostly in North and South America. However, during the last five years, widespread *X. fastidiosa* infections have been reported in a constrained area of the Apulia region (southern Italy), in olives trees suffering a severe disease, denoted as Olive Quick Decline Syndrome (OQDS). Because many xylem sap-feeding insects can function as vectors for the transmission of this exotic pathogen in EU, several research programs are ongoing to assess the role of candidate vectors in the spread of the infections. Initial investigations identified *Philaenus spumarius* (L.) as the predominant vector species in the olive orchards affected by the OQDS. Additional experiments have been carried out during 2016 and 2017 to assess the role of other species. More specifically, adults of the spittlebugs *Philaenus italosignus* Drosopolous and Remane, *Neophilaenus campestris* (Fallen) and of the planthopper *Latilica tunetana* (Matsumura) (Issidae) have been tested in transmission experiments to assess their ability to acquire the bacterium from infected olives and to infect different susceptible hosts (olives, almond, myrtle –leaf milkwort, periwinkle). Acquisition rates determined by testing individual insects in quantitative PCR assays, ranging from 5.6% in *N. campestris* to 22.2% in *P. italosignus*, whereas no acquisition was recorded for *L. tunetana*. Successful transmissions were detected in the recipient plants exposed to *P. italosignus* and *N. campestris*, whereas no trasmissions occurred with *L. tunetana*. The known vector *Philaenus spumarius* has been included in all the experiments for validation. The systematic surveys conducted in 2016 and 2017 provided further evidence on the population dynamics and seasonal abundance of the spittlebug populations in the olive groves.

## 1. Introduction

Since the discovery in 2013 of the first European outbreak of the insect-transmitted bacterium *Xylella fastidiosa* in southern Italy (Apulia region), different strains of the bacterium have been detected in Corsica and mainland France, Balearic Islands and mainland Spain [1,2,3], and in 2018 in Portugal and in central Italy (Tuscany) [4]. The presence of this bacterium in areas with favorable climatic conditions and the occurrence of competent insect vectors poses a serious threat to cropping systems and plants of landscape value. Due to the widespread infections discovered in the southern part of the Apulia region, in Corsica and Balearic Islands, in these areas, the bacterium is now declared established and containment measures are applied, including removal of infected sources and control of the vector populations [1].

Vectors of the bacterium are xylem-sap feeders belonging mainly to two *taxa* of the order Hemiptera: Cicadellidae Cicadellinae (commonly known as sharpshooters) and Cercopoidea (froghoppers or spittlebugs) [5,6,7]. No species-specific mechanisms mediating vector-*Xylella* recognition are known to occur; therefore, all xylem-sap feeders are potentially able to acquire and transmit the bacterium [8], although, with different efficiency depending on their host–plant preference, tissue preference, and other factors [9]. Bacterial cells attach to and multiply in the foregut of the insect vector [10] and, when acquired by adults, the bacterium is transmitted persistently in a noncirculative manner, with no transtadial or transovarial passage nor a latent period [10,11]. Sharpshooter leafhoppers are the primary vectors of the bacterium in America, but they are scantily present in Europe [12], where spittlebugs, froghoppers, and cicadas appear to be the predominant xylem-sap feeders [13]. Confirmations that spittlebugs play an important epidemiological role in the European agro-ecosystem were obtained with the identification of *Philaenus spumarius* (L.) as the first ascertained European vector responsible for the severe epidemic spread of the bacterium in the olive groves in the Apulian region [14,15,16]. More in-depth surveys in the olive groves of Apulia showed the presence of 15 different species of Auchenorrhyncha and indicated that *P. spumarius* was the most abundant species followed by *Neophilaenus campestris* (Fallen) (Aphrophoridae) and *Agalmatium flavescens* (Olivier) (Issidae) [17]. Although *P. spumarius* is the dominant Auchenorrhyncha species in the olive groves of the Apulia, in other areas, the faunistic composition of spittlebugs can be different. For example, in the Alicante Region of Spain, *N. campestris* has been reported as more abundant than *P. spumarius* [18] and, in specific biotopes, *Philaenus* species breeding on *Asphodelus* [19] can be relevant for the spread of *X. fastidiosa*, especially among non-cultivated plants, thus enlarging the natural reservoir of the bacterium.

This study aims to investigate role of two additional spittlebug species: *N. campestris*, widespread in Europe and in many other Mediterranean countries, and *Philaenus italosignus*, a species with limited distribution in Southern Italy and Sicily [20] as a vector of the Apulian strain of *X. fastidiosa*. Moreover, since feeding behavior of Issidae planthoppers has never been described, transmission tests included both aforementioned spittlebug species and *Latilica tunetana* (Matsumura) (Issidae), a planthopper that in our surveys was consistently found on olive canopies in the infected area of the Apulia region, and seldom found positive in PCR assays for *X. fastidiosa* presence. Experimental transmission tests were also supported by periodical field surveys in olive groves located in the infected area of Apulia to test for the presence of insects of these species carriers of *X. fastidiosa*, as well as to determine their seasonal fluctuations, i.e., understand their possible contribution in the epidemiology of the infections on olives.

## 2. Materials and Methods

### 2.1. Collection of Insects

Adults of *P. spumarius*, *P. italosignus*, and *N. campestris* were collected by sweeping net between June and September of two consecutive years (2016–2017) in different areas located in northern Apulia (*Xylella*-free area). More specifically, adults of *P. spumarius* (used as control for all experiments) were collected in olive groves, *P. italosignus* in cherry orchards and pine trees surrounding natural areas with abundance of *Asphodelus* spp., and adults of *N. campestris* collected from meadows and pine trees.

As for *L. tunetana*, in 2016, experiments were performed in the infected area, by collecting adults from the canopies of infected and symptomatic olive trees, which were then directly transferred on recipient plants for the transmission test. In 2017, transmission experiments with this planthopper were conducted similarly to those of the two spittlebugs, by collecting adults in olive groves located in the *Xylella*-free area and caging them on infected source plants.

Before setting each acquisition-transmission test, 15–20 specimens were selected for each species and assessed by qPCR [21] to confirm the absence of the bacterium.

### 2.2. Morphological and Molecular Characterization of Spittlebugs and Planthopper Species

Insects used for transmissions were identified at species level according to Drosopolous and Remane [20], Holzinger et al. [22], and Gnedzilov and Mazzoni [23]. Morphological identification was complemented by molecular characterization performed on selected representative specimens. For the two spittlebugs, a fragment of the mitochondrial gene *cytochrome c oxidase subunit I* (COI) was amplified by PCR using the primers C1-J-2195 and TL2-N-3014 [24]. For the planthopper, the primers targeting the COI gene were designed in this study (IS-fw 5′-GAA TTA TCA MAA CCA GGA TC-3′; IS-rev 5′-GGT CAC CTC CWC CTG AKG GAT C-3′), by selecting a conserved regions on the nucleotide alignment of sequences available in NCBI database, for *Latilica* spp., *Hysteropterum* spp. and *Agalmatium* spp.

Total DNA from the target insects was recovered using a standard CTAB-based extraction [25] upon a maceration step performed using a TissueLyser (Qiagen, City, US State abbrev. if applicable, Country). Five microliters of the purified DNA were used to set PCR reactions (50 µL) containing 300 nM of each primer and 2× DreamTaq Green PCR Master Mix (Thermo Fisher, Waltham, MA, USA) and using standard PCR conditions, with an annealing temperature of 50 °C for the primers C1-J-2195 and TL2-N-3014, and 58 °C for the primers IS-fw and IS-rev).

The recovered PCR amplicons were firstly purified using the Qiaquick PCR Purification kit (Qiagen, Valencia, CA, USA) and then sequenced. All the nucleotide sequences were deposited into NCBI GenBank.

### 2.3. Transmission Tests

Transmission experiments were performed in the demarcated infected area of Apulia. A total of 12 transmission experiments (6 experiments/year) were carried out between June 2016 and September 2017. Acquisition was performed by confining the insect specimens on branches of field-grown infected olive trees (cultivar Cellina di Nardò) located in the municipalities of Gallipoli and Parabita. The presence of the bacterium in these “donor” trees was assessed prior to each acquisition and transmission tests by sampling the young flushes. Individuals were caged in groups of 20–100 per branch for an acquisition access period (AAP) of 96 h. After the AAP, the insects were transferred onto receptor plants, in groups of five per plant, for an inoculation access period (IAP) of 96 h. The duration of AAP and IAP were chosen in order to maximize transmission efficiency. The IAP took place in a growth chamber at the regional nursery “Li Foggi” near Gallipoli (province of Lecce), at a temperature of 28–30 °C. In 2016, recipient plants consisted only of young olive seedlings, whose number varied from 17 to 30 for each insect species, depending on the number of insects that survived in the cages during the AAP. In 2017, myrtle-leaf milkwort and periwinkle were also included, as well as cherry and almond potted plants. These latter two were exposed to *P. spumarius* and *P. italosignus* only. For each insect species, the number of receptor plants varied from 5 to 16 for olives, from 5 to 21 for myrtle-leaf milkwort, and from 5 to 16 for periwinkle. Fifteen and 14 cherry plants were exposed to *P. spumarius* and *P. italosignus*, respectively, and similarly 15 and 7 of almond (Table 1).

For *L. tunetana* experiments in 2016, adults were collected on *X. fastidiosa* infected olive trees and transferred in groups of five onto healthy olive plants for an IAP of 96 h.

After the IAP, all the insects were stored in ethanol 70% at −20 °C. Plants were treated with a systemic insecticide and kept in an insect-proof screenhouse. At the end of each AAP and IAP experiment, data on survival of the insects were also recorded. From three to twelve months later, the test plants were analysed for *X. fastidiosa*.

### 2.4. Detection of X. fastidiosa in the Insects and Plants

All recipient plants except periwinkle were tested 12 months after each transmission experiment, whereas periwinkle plants were tested 3 months after the transmission test. Plant samples consisted of 4–8 leaves, from which the petioles and the midribs were recovered and used to purify the total DNA used to set the qPCR reactions [21]. All insect specimens collected from the cages after the IAP (either dead or alive individuals) were singly tested by qPCR [21] upon purifying the DNA from the excised heads using a CTAB-based extraction protocol [25]. The assays included three replicates of a 10-fold serial dilution of artificially spiked insect extracts with known bacterial concentrations ranging from 10^6^ to 10^1^ CFU/mL. An estimation of the bacterial concentration (CFU/head) in the positive specimens was extrapolated from the standard curve generated by quantitation cycle values of the spiked samples against the logarithm of their concentration.

### 2.5. Assessment of Spittlebugs Natural Carriers Collected in Olive Groves Located in the Xylella-Infected area of Apulia (Southern Italy)

Six different olive groves were selected in the demarcated “infected area” of Apulia and monitored every two weeks, for the presence and occurrence of spittlebugs (adults) from April to October of two consecutive years (2016 and 2017). Surveys were carried out by performing a pre-fixed number of sweeps on the ground vegetation, on the olive canopies and on border plants (mainly *Myrtus* sp., *Pistacia lentiscus and Cupressus* sp.) (Appendix A). More specifically, in each selected orchard, sampling units consisted of 20 olive trees with 10 sweeps for each olive canopy; 30 sites for sweeping on the ground vegetation with 4 sweeps for each sites; and 10 sites for the border plants with 4 sweeps for each sites.

Insects were separated according to the species, counted and kept in ethanol prior to be tested for the presence of *X. fastidiosa* by qPCR [21]. At each date of collection all the individuals collected, up to a maximum of 20 individuals per vegetation compartment (olive trees, ground vegetation, border plants), were subjected to diagnostic test and the % of *Xylella*-positive insects for each species estimated on a monthly basis.

### 2.6. Data Analyses

Statistical analysis was performed with the CoStat version 6.204 (CoHort Software). Statistical significance, for all data, was accepted for *p*-values <0.05 α-level. The chi-square test was used to identify significant differences in survivorship, acquisition, and transmission rates among the different spittlebugs and the planthopper used in our experiments. The chi-square test was also used to compare the frequency of infected specimens captured monthly. For the analysis of bacterial load in the insects, raw data of bacterial cells (CFU/head) were transformed into the logarithm, in order to stabilize the variance, and then compared using one-way ANOVA.

To study the fluctuation of spittlebugs on different vegetation compartments (olive canopies, ground vegetation and border plants), the total numbers of insects, captured monthly, were transformed in the √(10 + x) for the homogenization of variances and normality of the errors, being submitted to the variance analysis, considering the entire random delineation. Data on spittlebug population abundance in six different orchards located in the infected area were analyzed with one-way ANOVA and means compared with Tukey’s Test [26].

## 3. Results

### 3.1. Molecular Identification of the Insect Species Used in the Transmission Tests

The nucleotide sequences retrieved from the amplified COI genes on the DNA extracts recovered from specimens of the three spittlebugs showed high sequence identity with the sequences of the same species retrieved from the NCBI database, confirming the accurateness of the identification based on morphological characters. Accession numbers are available in NCBI GenBank under the accession numbers: MH165271 (*P. spumarius*), MH165272 (*N. campestris*) and MH165273 (*P. italosignus*).

Similarly, the sequences recovered for the COI genes of *L. tunetana* showed 99–100% of sequence identity with homologous sequences of *Latilica maculipes* (Melichar) and *Hysteropterum* spp., whereas they shared only 87% of nucleotide identity with sequences of *Agalmatium* spp. Accession numbers are available in NCBI GenBank under the accession numbers: MH249036.1, MH249037.1.

### 3.2. Insect Survival on the Different Plant Species

During both years, good survival rates were recorded (>69%) for the four insect species upon the 96 h of AAP on the olive branches (Figure 1). When the insects were transferred for the IAP, regardless the recipient plant species, a high percentage of spittlebugs survived compared to planthopper (Table 1 and Table 2).

After IAP, in 2016, a greater % of survival was recorded (χ^2^ = 90.32, *p* < 0.001) on olive plants for *P. spumarius* (88.23%) and *P. italosignus* (83.33%) compared to *N. campestris* (58.82%) and *L. tunetana* (28.77%).

In the experiments carried out in 2017, analysis of the insects that survived the IAP on olive seedlings confirmed different survival rates between spittlebugs and the planthopper *L. tunetana* (χ^2^ = 34.76, *p* < 0.001), whereas no significant difference was recorded among the spittlebug species (χ^2^ = 5.84, *p* = 0.054). High survival rates (>81%) were recorded for *P. spumarius* and *P. italosignus* during the IAP on almond and on cherry, and all three spittlebug species showed similar high survival rates on myrtle-leaf milkwort (>81%) (χ^2^ = 2.25, *p* = 0.52). On periwinkle, different survival rates (χ^2^ = 10.34, *p* = 0.0056) were recorded, with the highest survival rates for *P. spumarius* (90.47%) compared to *N. campestris* (50.9%) and *L. tunetana* (52.5%).

### 3.3. Acquisition Rates

In the transmission experiments conducted in 2016, no differences were observed for the acquisition rates (four days AAP) among spittlebugs (χ^2^ = 3.04; *p* = 0.218), with an acquisition rate of 20.17% for *P. spumarius*, 16.66% for *P. italosignus* and 10.30% for *N. campestris* (Table 1). Two out of 73 (2.74%) of the individuals of *L. tunetana* captured on olive canopies in the infected area tested positive in qPCR assays for the presence of *X. fastidiosa*.

In 2017, the acquisition rates ranged from 24% for *P. italosignus* to 15.6% for *P. spumarius,* and to 5.6% for *N. campestris*, being different among the three spittlebugs (χ^2^ = 22.92, *p* < 0.001). *L. tunetana* did not acquire the bacterium. Differences in acquisition rate were observed comparing *P. spumarius* versus *P. italosignus* (χ^2^ = 4.39; *p* = 0.036), *P. spumarius* versus *N. campestris* (χ^2^ = 9.94; *p* = 0.002), and *P. italosignus* versus *N. campestris* (χ^2^ = 23.21; *p* = <0.001) (Table 2, Appendix A).

### 3.4. Transmission Rates

In 2016, the bacterium was successfully transmitted to olive seedlings by all the three spittlebug species, whereas no transmission occurred with adults of *L. tunetana* collected from infected olive trees (Table 1). *Philaenus spumarius* proved to be a more efficient vector compared to *P. italosignus* and to *N. campestris* (χ^2^ = 6.588, *p* = 0.037).

Estimated probability of transmission by single vectors (E) [27], was 0.061 for *P. spumarius*, 0.01 for *N. campestris* and 0.014 for *P. italosignus*.

In 2017, the three spittlebugs transmitted *X. fastidiosa* to myrtle-leaf milkwort, with a rate of 60% for *P. spumarius*, 33.33% for *P. italosignus* and 9.5% for *N. campestris* (Table 2). *P. spumarius* and *P. italosignus* transmitted the bacterium to olive, with transmission rates of 31.3% and 7.69%±, respectively. No transmission to olive occurred with *N. campestris. P. italosignus* was the only species that transmitted the bacterium to cherry (one plant out of 14). Neither *P. spumarius* nor *P. italosignus* transmitted the bacterium to almond.

*L. tunetana* did not transmit the bacterium to any of the 25 receptor plants (olive, myrtle-leaf milkwort, periwinkle) exposed to the insects.

Estimated infectivity of single vectors (E) was higher for *P. spumarius* than for other spittlebugs on olive seedlings, myrtle-leaf milkwort and periwinkle.

### 3.5. Estimation of the Bacterial Load in the Vectors

Quantitative PCR assays allowed for estimating the bacterial load in the three spittlebug species. Tests conducted in 2016 yielded the following average values of CFU/head: 820.7 ± 316.4 SE, 2044.6 ± 1090.7 SE and 183.0 ± 56.7 SE, for *P. spumarius*, *P. italosignus* and *N. campestris*, respectively, with no significant differences in the bacterial load estimated in the three spittlebugs (*p* = 0.328).

In 2017, the bacterial population measured in *P. italosignus* (1,260.1 ± 395.3 SE) was significantly higher than in *P. spumarius* (305 ± 94.0 SE) and *N. campestris* (142.5 ± 71.4 SE) (*p* = 0.008).

### 3.6. Population Density and Natural Infectivity of Spittlebugs Collected in Olive Groves Located in the Infected Area of Apulia

Although differences in the population density were recorded between *P. spumarius* and *N. campestris*, a similar trend was observed for both species during the two consecutive years (Figure 2, Appendix A). Adults of both species appeared in late April when they were collected mainly on the ground vegetation, reaching in general a peak in May and then a constant reduction until the end of the season in October.

*P. spumarius*. Regardless of the vegetation compartment, a peak of adults was recorded in May (Figure 2, Appendix A) reaching the following average values: On ground vegetation 4.4 specimens/sample unit in 2016 and 5.4 specimens/sample unit in 2017; on olive canopies 4.5 specimens/sample unit in 2016 and 2.6 specimens/sample unit in 2017; on border plants 5.4 specimens/sample unit in 2016 and 6.7 specimens/sample unit in 2017. Following this peak, in June, the population started to decrease, particularly on the ground vegetation (drying during the summer), with the majority of insects being collected on border plants. In 2016, the number of adults collected on the ground vegetation started to increase again in late August, following a rainfall period that promoted the germination of several weeds species; conversely, in 2017, when the summer season was very dry, a notable increase in the number of individuals collected on ground vegetation was registered in October (F = 68.48; d = 5; *p* < 0.0001). The overall number of adults collected from olive trees in 2017 was lower than in 2016, probably because of desiccation of olive foliage as a consequence of *X. fastidiosa* infection.

For both years, the first *Xylella*-positive individuals of *P. spumarius* were detected in May, when adults peaked on olive trees (Figure 3, Appendix A), and a peak of infected specimens was always detected in June (reaching 50%), soon after the movement of the adults from the ground vegetation to the olive canopies. In 2016, the incidence of positive *P. spumarius* captured on olive trees remained substantially constant from June to October. In 2017, after the peak of positive samples recorded in June, the incidence of infected specimens decreased rapidly from July to August, and slightly increased at the end of the season (20.5%) in September. On border plants, upon the first detection of positive individuals in May (2016: 6.0%; 2017: 4.0%), a trend with a continuous increase was recorded afterwards, with a peak at the end of the season (2016: 46.2%; 2017: 23.2%). Few *P. spumarius* adults collected from the ground vegetation tested positive for the bacterium in May (2016: 3.8%; 2017: 2.5%); then, their proportion sharply increased in August (2016) and at the end of September/October (2017), when adults move from the olive canopies to the newly emerged weeds.

*N. campestris.* During both years, the population densities were consistently lower than those recorded for *P. spumarius*, regardless of the monitored compartment (2016: F = 57.28, df = 5, *p* < 0.0001; 2017: F = 244.73, df = 5, *p* < 0.0001). Similarly, the occurrence of *Xf*-positive individuals of *N. campestris* was lower than *P. spumarius* on olive canopies (2016: χ^2^ = 69.10; *p* < 0.0001; 2017: χ^2^ = 16.5; *p* < 0.0001), border plants (2016: χ^2^ = 24.69; *p* < 0.0001); 2017: χ^2^ = 17.0; *p* < 0.0001) and weeds in 2016: χ^2^ = 34.65; *p* < 0.0001.

## 4. Discussion

Knowledge on the vector species responsible for the spread of *X. fastidiosa* is critical for understanding the epidemiology and for developing adequate control measures to reduce the impact of the diseases caused by this vector-borne bacterium. We carried out transmission tests on olives, cherry, almond, myrtle-leaf milkwort, and periwinkle plants, using three spittlebugs, *P. spumarius*, *P. italosignus*, and *N. campestris,* and the planthopper *L. tunetana*. The experiments herein described showed that, besides *P. spumarius*, two additional spittlebug species are competent vectors of the strain of *X. fastidiosa* subsp. *pauca* ST53 associated with the severe epidemics in Apulia (southern Italy). Conversely, the negative results obtained with *L. tunetana*, a planthopper species that we found to be abundant on the olive canopies, represent an indirect experimental confirmation of the phloem-feeding behavior of this insect and a further confirmation that phloem-feeders may occasionally ingest *X. fastidiosa* cells, thus testing *Xylella*-positive, but they are unable to transmit [17,28,29], as demonstrated in our experiments. It has to be noted that, during the transmission experiments, lower survival rates were recorded on source and recipient plants for *L. tunetana* compared to the spittlebugs. However, even if the low survival rates could impair the transmission efficiency, it should be considered that, in previous experiments, high mortalities were recorded for *P. spumarius* on oleander plants, but still high transmission rates were obtained [15].

With regard to the host plants, the overall results showed that these three spittlebugs were able to transmit the bacterium to olive and myrtle-leaf milkwort. On the contrary, insect transmission to cherry was very inefficient (only one plant exposed to *P. italosignus* became infected), and, under our semi-field experimental conditions, neither *P. spumarius* nor *P. italosignus* were able to transmit to almond. This is in line with the field observations, indicating that, in the olive infected area, few almond and cherry plants are infected. Actually, in a recent survey in the heavily infected area, while 100% of the olives sampled in different foci tested positive, the rate of positive almond trees was as low as 20–25% (Saponari, unpublished data).

*P. italosignus* was rarely found in the olive groves surveyed in the past three years in the infected area where olive is the predominant crop, while its presence has been recorded in the central and northern part of the Apulia region, where a larger diversity exists in the agricultural crops, and, in the event of an expansion of the pathogen toward the north of the Apulia region, this vector species may play an important role in transmitting the bacterium to different crop species. With regard to the host preference of *P. italosignus*, while it is well known that nymphs develop exclusively on asphodel (*Asphodelus* spp.), this also serves as a host plant for oviposition [19]; limited information is available for the adults. Given its competence for *X. fastidiosa* transmission, developing knowledge on the biology, population dynamics and abundance will be crucial for an effective pest management strategy in case of the spreading of *X. fastidiosa* in agro-ecosystems where this species is present [30].

Juveniles and adults of *N. campestris* are widely polyphagous, although they have a different host preference and ecological niches than *P. spumarius*. Indeed, previous surveys carried out on the ground vegetation of the Apulian olive groves indicated that the population density of the nymphs of *N. campestris* was about 10 times lower than *P. spumarius* [31]. This is in line with the lower population density monitored during our surveys. Indeed, similar results were recently obtained in Spain by Morente et al. [32], where adults of *N. campestris* were never caught on olive canopies, and in olive groves of the Bari province of Apulia, where population densities of *N. campestris* were consistently lower than those of *P. spumarius* [33].

It is worth noting that, while the two species of *Philaenus* seem to transmit *X. fastidiosa* with similar efficiencies, *N. campestris* was always less efficient than both *Philaenus* species in transmitting *X. fastidiosa* to olive and myrtle-leaf milkwort.

The systematic surveys conducted in 2016 and 2017 provided further evidence on the population dynamics and seasonal abundance of the *P. spumarius* and *N. campestris* in the olive groves and seasonal prevalence of *Xylella*-positive specimens of these two spittlebug species. *P. spumarius* is clearly the most abundant Auchenorrhyncha in the olive groves, confirming the preliminary data collected during the surveys carried in the infected area at the early stage of the epidemics [14]. Under the monitored field conditions, *Xylella*-positive specimens of *P. spumarius* started to be detected soon after the emergence of the adults in early May and their incidence increased throughout the season. This indicates that, as soon the adults start to move to the olive canopies, they are able to acquire the bacterium. For an effective and sustainable control strategies, efforts should be made in the early phase of the adult season, with the attempt to reduce as much as possible the number of adults visiting the infected olive canopies. An increasing proportion of *Xylella*-positive specimens was recorded over the season, regardless of the vegetation compartment. If we consider that border plants present in our plots were non-hosts of the bacterium and that *X. fastidiosa* infections on herbaceous species (ground vegetation) are rare and have been detected only in autumn, the infected spittlebugs detected on border plants and ground vegetation throughout the season are most probably the result of a continuous dispersion of the vector insects from the olive canopies, where they acquire the bacterium, to other suitable hosts present in the orchards.

On the other hand, our results clearly indicate that the occurrence of *Xylella*-positive individuals of *N. campestris* is significantly lower than *P. spumarius,* and generally limited to the initial part of the season (May). This is likely due to the non-preference of this species for olive [33], which is the main host of *X. fastidiosa* in the area. Such evidence is important for pest risk assessment and vector management in different agro-ecosystems, considering that both species are quite common and widespread in several European countries, including those where *X. fastidiosa* has been recently detected.

Regarding the seasonal fluctuation of the populations, during the two years, similar trends were recorded for both spittlebugs, with a peak in May soon after the emergence of the adults. Later in the season, a decrease of the populations is observed for both *P. spumarius* and *N. campestris* on the olive canopies, with the complete disappearance of the latter species. At the same time, an increase of population density is observed for the border plants, where most probably they find tender shoots for feeding and more suitable microclimatic conditions. The trends observed in the Apulian olive groves are quite similar to those recorded in Spain by Morente et al. [32], where a similar migration pattern occurs, although, in our conditions, *P. spumarius* never disappear from the olive canopies, indicating that, under the Apulian conditions, there is a continuous risk of bacterial infections in olives over the spittlebug adult season.

## 5. Conclusions

The vector-mediated transmission experiments conducted over a two-year period showed that, besides *P. spumarius*, two additional spittlebug species are competent vectors of the strain of *X. fastidiosa* subsp. *pauca* ST53 associated with the severe epidemics in Apulia (southern Italy). Successful bacterial transmissions were detected in the recipient plants exposed to *P. italosignus* and *N. campestris.*

The occurrence of different insect species able to act as vectors and characterized by different host–plant preference/selection can strongly influence the range of plant species that are exposed to *X. fastidiosa*, thus making possible new and multiple epidemiological cycles among wild species and between these latter and cultivated plants.

## Figures and Tables

**Figure 1 insects-10-00324-f001:**
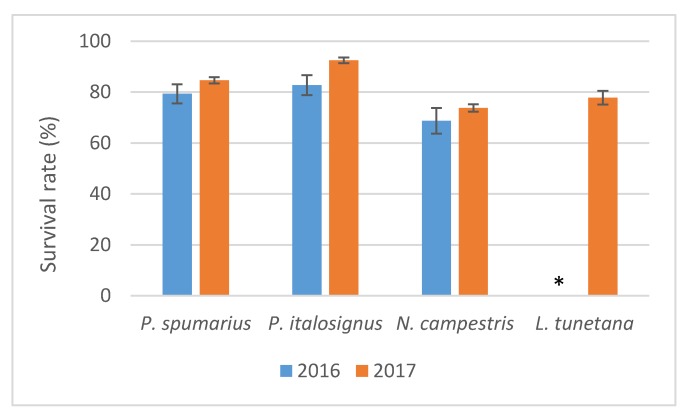
Survival (%) recorded in 2016 and 2017 upon four days of acquisition access period on olives for *Philaenus spumarius*, *P. italosignus*, *Neophilaenus campestris* and *Latilica tunetana.* * not done in 2016.

**Figure 2 insects-10-00324-f002:**
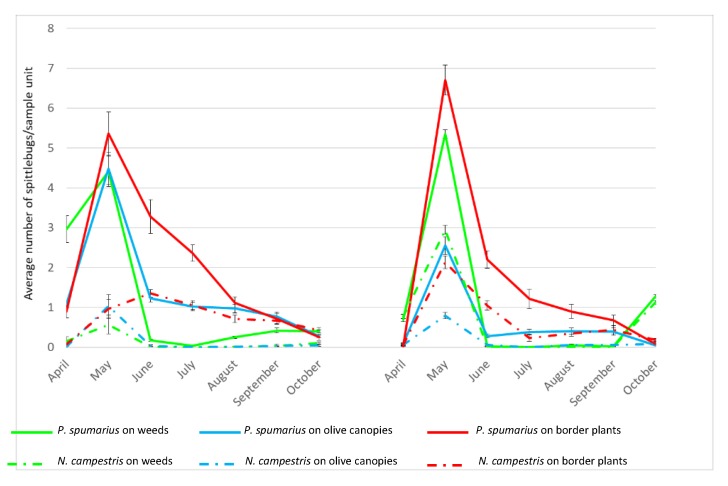
Population abundance of *Philaenus spumarius* and *Neophilaenus campestris* in the infected olive groves monitored in 2016 and 2017.

**Figure 3 insects-10-00324-f003:**
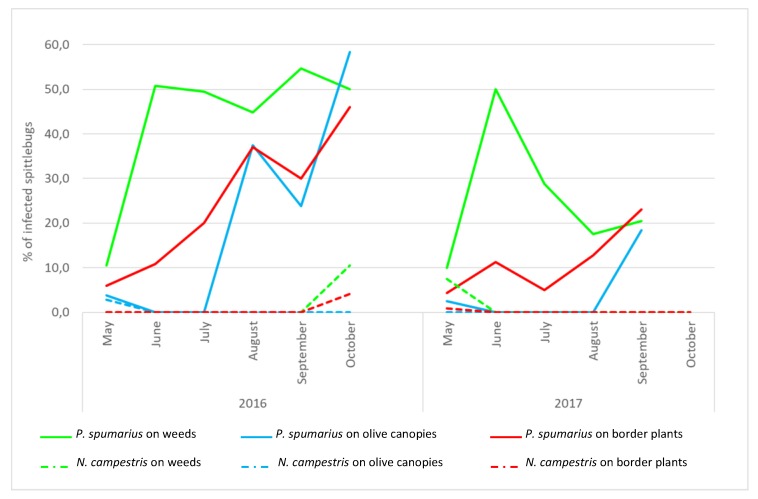
Occurrence of *Xylella*-positive individuals of *Philaenus spumarius* and *Neophilaenus campestris* on infected olive canopies, ground vegetation and border plants, over the two years of surveys.

**Table 1 insects-10-00324-t001:** Data collected from the six transmission experiments carried out in 2016 using specimens of different insect species caged for an acquisition access period of 96 h on *Xylella fastidiosa* infected olive trees. Number of insects and recipient olive plants used for the inoculation access period (IAP) are indicated.

Insect Species	N. of Insects		N. of Recipient Plants		E ^b^
	Used for IAP	Alive after IAP (% ^a^)	Total	Infected after IAP (%)	
**Olive (*Olea europaea*)**					
***Philaenus spumarius***	150	105 (88.23)	30	8 (26.66)	0.061
***Neophilaenus campestris***	100	40 (58.82)	20	1 (5)	0.010
***Philaenus italosignus***	145	100 (83.33)	29	2 (6.89)	0.014
***Latilica tunetana***	85	21 (28.76)	17	0	0

^a^ The percentage is calculated based on the total insects found in the cages after the IAP. ^b^ Formula of Swallow (1985): Estimated probability of transmission for single insect E = 1 − (1 − p)1/k, in which p = proportion of infected plants and K = number of individuals used per tested plants.

**Table 2 insects-10-00324-t002:** Data collected from the six transmission experiments carried out in 2017 using specimens of different insect species caged for an acquisition access period of 96 h on *Xylella fastidiosa* infected olive trees. Number of insects and recipient plants of the different host species used for the inoculation access period (IAP) are indicated.

Insect Species	N. of Insects		N. of Recipient Plants		E ^b^
	Used for IAP	Alive after IAP (% ^a^)	Total	Infected after IAP (%)	
**Cherry (*Prunus avium*)**					
***Philaenus spumarius***	75	51 (80.95)	15	0	0
***P. italosignus***	70	53 (79.10)	14	1 (7.14)	0.015
**Almond (*Prunus dulcis*)**					
***P. spumarius***	75	56 (88.88)	15	0	0
***P. italosignus***	35	33 (94.28)	7	0	0
**Olive (*Olea europaea*)**					
***P. spumarius***	80	62 (93.93)	16	5 (31.25)	0.072
***Neophilaenus campestris***	55	39 (86.66)	11	0	0
***P. italosignus***	65	46 (79.31)	13	1 (7.69)	0.016
***Latilica tunetana***	25	9 (39.13)	5	0	0
**Myrtle-leaf milkwort (*Polygala myrtifolia*)**					
***P. spumarius***	25	22 (91.67)	5	3 (60)	0.167
***N. campestris***	105	66 (81.48)	21	2 (9.5)	0.020
***P. italosignus***	45	34 (89.47)	9	3 (33.33)	0.078
***L. tunetana***	15	6 (85.7)	10	0	0
**Periwinkle (*Catharanthus roseus*)**					
***P. spumarius***	25	19 (90.47)	5	3 (60)	0.167
***N. campestris***	80	26 (50.98)	16	0	0
***L. tunetana***	50	21 (52.5)	10	0	0

^a^ The percentage is calculated based on the total insects found in the cages after the IAP. ^b^ Formula of Swallow (1985): Estimated probability of transmission for single insect E = 1 − (1 − p)1/k, in which p = proportion of infected plants and K = number of individuals used per tested plants.

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
