# Peer review of "Transmission of Xylella fastidiosa Subspecies Pauca Sequence Type 53 by Different Insect Species"

_insects, 2019, doi:10.3390/insects10100324_

Round 1
Reviewer 1 Report
This manuscript investigates “Transmission of Xylella fastidiosa Subspecies Pauca Sequence Type 53 by Different Insect Species". The experimental set up of this study appears to be well-designed and the data collected carefully. In results section, Duncan's multiple range test is not rigorous, I suggest replace by Tukey HSD test. I think that this manuscript requires substantial rewriting to make its introduction section clearer and more readily interpretable to the reader. The authors provide knowledge in this manuscript and can be of great interest to the journal. Based on the comments above reported, my opinion is that this manuscript may be suitable for printing on this journal.
A few points:
Lines 22-26: Sentence is Very extensive and confuse. Please, separate into two sentences
Line 30: Sentence starting “Philaenus spumarius has been included…”
Lines 33: Keywords should be in alphabetic order
Line 50: Place “,” after preference
Line 51: Delete “the”
Line 57: For better interpretation, I suggest including the taxon (family) for all species mentioned in the manuscript: Neophilaenus campestris (Fallen) (Aphrophoridae)
Line 61: …and Agalmatium flavescens (Olivier) (Issidae)…
Line 68: …species:N. campestris, widespread…
Line 74: Delete “(Cavalieri, unpublished data)”
Line 83: Sentence starting “Adults of P. spumarius…”
Line 84: Delete “mainly”
Line 87: Delete “directly”
Line 95: …Remane [20], Holzinger…
Line 112: …of 96 h.
Line 114: …of 96 h.
Line 124: Delete “directly”
Line 124: …of 96 h.
Table 1: Change “P. italosignus” by “Philaenus italosignus”
Line 146: CFU/mL.
Line 159: The citation (Harper et al., 2010) should be included in references according to the journal style. Please check citation No. 21
Line 166: Place “,” after acquisition
Line 176: Replace “Duncan’s Multiple Range Test” by “Tukey HSD test”. Also, check the results
Line 191: …96 h
Figure 1, 2 and 3: Scientific names should be in italic
Line 209: Delete “statistically significant”
Line 215: Delete “significantly”
Line 220: …96 h
Line 237: …31.3%± and…
Line 238: Place “.” after respectively
Lines 247, 248, 250 and 251: What does SE mean?
Line 292: Delete “significantly”
Line 238: Place “,” after milkwort
Line 321: Delete”(Dongiovanni, unpublished data)”
Lines 355-356: Delete”(Dongiovanni, unpublished data)”
References: In all references, scientific names should be in italic
Reviewer 2 Report
To the authors:
In the present study, titled as ‘Transmission of Xylella fastidiosa Subspecies Pauca 2 Sequence Type 53 by Different Insect Species’, the authors examined the acquisition and transmission of X. fastidiosa by two species of spittlebugs and one species of planthopper in different plant species. In summary, the work is not well structured. However, the topic and the results are worth it to be published. This study includes the logical steps commonly investigated in acquisition and transmission studies. I encourage the authors to improve the structure of the experimental design and results before publication. For example, i) the manuscript showed results (Table 1) in materials and methods, move table 1 to the proper section. In point 2.1. Collection of insects, need to be divided in two parts, 2.1. collection of insects and 2.2. morphological and molecular characterization of spittlebugs and planthopper species. ii) Molecular biology protocols are missing in the manuscript (DNA extractions and PCR conditions), when they are critical in this research. iii) A supplementary figure showing the areas chosen for insect collection in olive groves and vegetation compartment might simplify the spatial and visual distribution for manuscript readers. iv) Add a plot showing the evaluation of the bacterial load in the vectors.
As a reviewer, I appreciate the time and effort that was performed into the preparation of your article, but this study needs to be improved before publication.
Questions to the authors:
How the PCR amplicons were purified and sequenced? Please, add these protocols to the manuscript Which protocol-reagents were used for PCR amplification? The acquisition and transmission times were chosen using which criteria? Why the PCR results were extrapolated from the standard curve? When dilutions can be achieved. The transformations performed to homogenize the variances, for example √ (10 +x), were considered and/or following any statistic manuscript or book? As an author never did a transformation more than the logarithmic, so this question arises from my inexperience in this matter.
Suggestions
Line 15-18. Most of the diseases associated to the insect-transmitted bacterium Xylella fastidiosa have been described in North and South America, but in the last five years widespread infections were reported in a restricted area of the Apulia region (southern Italy), on olives suffering a severe disease denoted “olive quick decline syndrome – OQDS”.
It should be
Diseases associated with Xylella fastidiosa have been described mostly North and South America. But, during the last five years, widespread X. fastidiosa infections have been reported in a constrained area of the Apulia region (southern Italy), in olives trees suffering a severe disease, denoted as Olive Quick Decline Syndrome (OQDS).
Line 36. After the discovery in 2013…
Replaced. After with Since
Line 39. The presence of this highly polyphagous bacterium
Eliminate ‘highly polyphagous’ from the sentence
Line 67. In this perspective, our study aimed at investigating the role of..
Change: This study aims to investigate the role ….
Line 95. Fix Drosopolous & Remane reference.
Line 97. cytochrome c oxidase subunit I should be italic
Line 146. ranging from 10^6 to 10 CFU/ml.
Should be … 106 to 101 CFU/mL
Line 192-193. These lines need to be rewritten.
a high percentage of spittlebugs survived, while a lower percentage of planthopper did it (Table 1 and 2).
a high percentage of spittlebugs survived compared to planthopper (Table 1 and 2).
Line 238. Add a period after respectively.
Reviewer 3 Report
The work submitted by Cavalieri et al. is a timely and significant contribution to the field. I enjoyed reading the manuscript and only have the following minor points that the authors may wish to consider:
L29-30: '...whereas L. tunetana did not transmit': please check this, you may need to rephrase. L125-126: 'Plants were treated....': please state the reason why you did this. 'acquisition access period': consider changing to 'acquisition period', the same is 'inoculation period' L140: 'Plant samples...': do you mean 'biological replicates'? L331: ...although they have...'Author Response
Please see the attachment

Round 2
Reviewer 1 Report
The manuscript “Transmission of Xylella fastidiosa Subspecies Pauca Sequence Type 53 by Different Insect Species” has been improved and all my questions were taken into account. I recommend the publication in “Insects”.